# The Influence of Hydropower and Coal Consumption on Greenhouse Gas Emissions: A Comparison between China and India

Ugur Korkut Pata [1] and Amit Kumar [2,3,*]

1 Department of Economics, Osmaniye Korkut Ata University, Osmaniye 80000, Turkey; korkutpata@osmaniye.edu.tr
2 School of Hydrology and Water Resources, Nanjing University of Information Science and Technology, Nanjing 210044, China
3 Department of Hydro and Renewable Energy, Indian Institute of Technology Roorkee, Uttarakhand 247667, India
* Correspondence: amitkdah@nuist.edu.cn

**Abstract:** This study mainly aims to investigate carbon status according to the Pollution Haven Hypothesis (PHH) in developing countries such as India and China based on annual time series data from 1980 to 2016. The recently developed bootstrap autoregressive distributed lag procedure is applied to observe the long-run effect of FDI, hydropower, and coal-based fossil fuel consumption on three repressive measures of carbon emissions. The empirical results of the analysis show that hydropower and coal consumption lead to an upsurge in carbon emissions and the size of the carbon footprint in China. Similarly, Chinese FDI increases the carbon footprint. Moreover, Indian FDI and coal consumption accelerate carbon emissions while hydropower has no impact on environmental degradation. These results suggest that the PHH exists in China and India and that the validity of the PHH varies according to differing carbon indicators. Based on the empirical results, effective policy practices can be implemented by replacing coal and hydropower with more effective renewable energy sources and allowing foreign investors to pursue environmental concerns in the fight against environmental degradation.

**Keywords:** hydropower; coal consumption; PHH; $CO_2$ emissions; carbon footprint





## 1. Introduction

Since the 1990s, energy demand has increased worldwide due to rapid industrialization, urbanization, and economic growth. With this high-energy consumption, especially in developing countries (i.e., India, China), a significant amount of carbon dioxide ($CO_2$) is emitted, which poses a high risk to the environment and further affects climate change. Therefore, promoting renewable energy and implementing energy conservation measures play an important role in providing long-term solutions for a sustainable environment and socio-economic development. The use of renewable energy sources (RES) also improves the economic performance of a country and ensures energy security. In this regard, developed and developing economies are giving more priority to renewable-based energy sources to mitigate $CO_2$ emissions and reduce their dependence on fossil fuel sources (e.g., coal-fired power plants). Among RES, hydropower has emerged as the oldest renewable energy source that is not capital-intensive, produces less $CO_2$ emissions compared to fossil fuel sources, and provides direct or indirect employment opportunities. As a low-carbon, low installation cost, reliable, and clean source of energy, hydropower is playing an important role in carbon reduction and further helps in mitigating future climate change. Compared with highly volatile and non-steady renewable energy, flexible hydropower can be highly supportive to secure reliable grid operations [1]. Hydropower can meet energy demand and significantly reduce $CO_2$ emissions compared to coal-fired power plants [2]. For

this reason, the development of hydropower has increased rapidly worldwide in recent decades. With the large investments (~23.3 billion USD) in hydropower projects in 2016, the global-installed hydropower capacity reached 1246 GW, while the estimated generation capacity was 4102 GW [3].

China and India ranked first and sixth in global hydropower generation [4]. After the pre-industrial era, the installed capacity of hydropower in China increased from 11.74 GW to 330 GW in 2011, which was 25% of the global capacity and was expected to increase to 350 GW by 2020 [5]. In 2018, China and India were the largest emitters of terrestrial $CO_2$ emissions from fossil fuels globally, with 27.5 and 7.26 metric tonnes, respectively [6,7]. According to the IEA report [8], India's primary energy demand increased by 4% and reached over 35 Mtoe in 2017–2018 [8]. Moreover, coal-based energy sources contribute the largest share to energy demand and act as the first major contributor to energy demand growth. For these reasons, coal and hydropower consumption are likely to be the determinants of $CO_2$ emissions in China and India.

Foreign direct investment (FDI), i.e., foreign companies investing directly in the host country, is another important factor affecting $CO_2$ emissions. FDI is becoming a broad subject of interest and research in terms of job creation, access of local firms to international markets, development of managerial skills, increase in efficiency, and provision of technology transfer to local firms [9]. Thus, FDI inflows have played a fundamental role in economic development and also in the environment of developing countries. Although FDI has a positive impact on economic growth, it can have different effects on environmental quality through three channels: (i) scale effect, (ii) technique effect, and (iii) composition effect [10,11]. The scale effect implies that FDI increases pollution by augmenting the size of the country's economy [12], the composition effect states that FDI can reduce or increase pollution by changing the economic pattern [10], while the technique effect stipulates that foreign firms can adopt more environmentally friendly technology and improve the environment through emission-reduction approaches. In the literature, the pollution-reducing capabilities of FDI are referred to as the pollution halo effect. Theoretically, we can say that FDI inflows can have both positive and negative effects on the environment.

As environmental controls increase production costs, dirty industries and entrepreneurs shift the production process to less-developed regions where environmental policies are laxer [13]. In other words, with the rapid increase in competition for FDI, industries that cause high levels of pollution in developed countries are likely to move to developing countries due to the strict policy regulations and high costs of mitigation in developed countries [14]. Thus, some developing countries are turning into pollution slums. This phenomenon is known in the environmental literature as the pollution haven hypothesis (PHH). In the case of the PHH, economic expansion caused by FDI inflows into dirty sectors increases the level of pollution [15]. The presence of the PHH mainly depends on the quality and volume of FDI inflows [16].

Although fund flows are declining in many OECD countries, increasing FDI inflows to developing countries in Southeast Asia (including China and India) account for 39% of global inflows in 2018, leading to an increase in annual $CO_2$ emissions [17]. China and India are among the top three countries with the highest $CO_2$ emissions. The increasing environmental pressure in the two countries has led to severe problems worldwide (climate change, ozone depletion, global warming, etc.). In particular, the economic reforms implemented in the countries in the 1990s led to liberalization and high FDI inflows, which accelerated environmental degradation. Therefore, determining the influence of FDI and renewable and non-renewable energy sources on environmental pollution in these countries is an important issue for environmental policies.

To date, there has been no comparative analysis on the environmental impact of hydropower and coal consumption for China and India. These two countries are responsible for more than 35% of the global population. In addition, China and India are the two largest coal consumers and are among the top six largest hydropower-consuming countries in the world [18]. This high population and coal consumption cause various

environmental problems such as increased atmospheric $CO_2$, negative effects on human health, an ecological risk to the population, etc. China and India are two of the three highest $CO_2$-emitting countries in the world, and therefore, reducing carbon emissions is important both nationally and globally. In this regard, this study aims to examine the long-run impacts of hydropower, coal consumption, and FDI on carbon-based pollution for India and China using an advanced empirical model. To this end, the study addresses three research questions: (i) Is the PHH valid in China and India in terms of carbon footprint and $CO_2$ emissions? (ii) Is hydropower a solution to reduce environmental pollution? (iii) To what extent does coal consumption increase pollution? Therefore, the results of this study could be useful for developing countries to take upfront measures to reduce pollution by implementing appropriate environmental policies.

The remainder of the paper is organised as follows: Section 2 reviews the findings of studies testing the PHH for China and India. Section 3 defines different models for three pollution indicators used in the study and explains the application and advantages of the bootstrap ARDL method. Section 4 evaluates the results of the unit root test, cointegration relations, and the short and long-run estimates. Section 5 discusses the empirical findings and limitations of this study. Finally, the last section presents policy recommendations and conclusions.

## 2. Literature Review

Over the past decades, empirical models of environmental economics have been used to analyse the relationships between FDI, renewable energy consumption, and carbon pollution. However, the theoretical and empirical evidence on the relationships between environmental degradation and FDI inflows is inconclusive [10] and still highly controversial.

Recently, a number of studies have investigated the validity of the PHH using various panel-data techniques: Shahbaz et al. [19] for the $CO_2$ emissions of 99 countries, Sapkota and Bastola [20] for the $CO_2$ emissions of 14 Latin American countries, Balsalobre-Lorente et al. [21] for the ecological footprint of Mexico, Indonesia, Nigeria, and Turkey, Mert et al. [22] for $CO_2$ emissions of 26 European countries, and Sarkodie and Strezov [23] for $CO_2$ emissions and non-$CO_2$ greenhouse gas emissions of China, India, Indonesia, Iran, and South Africa. These studies validated the PHH, implying that FDI inflows lead to a reduction in environmental quality. However, the following studies reported findings in the opposite direction: Mert and Boluk [24] for $CO_2$ emissions of 21 Kyoto countries, Albulescu et al. [25] for $CO_2$ emissions of 14 Latin American countries, Destek and Okumus [26] for the ecological footprint of 10 newly industrialized countries, Shao et al. [27] for $CO_2$ emissions of 9 developing countries, and Salehnia et al. [28] for $CO_2$ emissions of 14 MENA countries. As can be seen in the above studies, there is no consensus on the validity of PHH.

Numerous studies in the literature examine the PHH using time series for a single country. In this context, the empirical results to date regarding the PHH for China and India are mixed. China and India are among the top 10 countries that received the most FDI inflows in 2018 [17]. China is generally cited as a good example of FDI inflows compared to $CO_2$ emissions and holds the top position in FDI inflows globally. The Chinese economy has grown continuously at or above 8% over the past three decades, resulting in high industrial emissions. With the rapid increase in economic activities followed by high-energy consumption, China is becoming the world's largest emitter of $CO_2$, leading to a high vulnerability to climate change in the future if strategic mitigation plans are not implemented in the coming decades [29]. Recently, China has introduced several green investment incentives as well as tax credits and investment allowances for pollution abatement equipment to minimize carbon pollution [30]. Based on all these facts, China and India are good examples to study FDI environmental pollution comparatively and learn from developing countries. Table 1 summarizes the outcomes reached by the selected studies on the PHH for China and India. As can be seen in the table, there is no consensus on the validity of the PHH for China and India. Four of the nine studies in the literature

argue that the PHH is valid in China, while the other five discuss the opposite results. For India, one out of four studies does not support the PHH. It is expected that the present study can help in providing insights for a sustainable hydropower future so that both countries can achieve their emissions target and contribute to carbon reduction and global warming impacts.

**Table 1.** Literature review on empirical findings of the PHH for China and India.

| Author(s) | Time Period | Country | Methodology | Results |
|---|---|---|---|---|
| He [31] | 1994–2011 | China's 29 provinces | GMM | PHH (✓) FDI increases $SO_2$ emissions |
| Acharyya [15] | 1980–2003 | India | OLS | PHH (✓) FDI increases $CO_2$ emissions |
| Zhang and Zhou [12] | 1995–2010 | China | STIRPAT | No PHH for $CO_2$ emissions. |
| Liu et al. [32] | 2002–2012 | 112 Chinese cities | OLS, GMM, FE | No PHH. FDI reduces $SO_2$ and soot emissions. |
| Sun et al. [16] | 1980–2012 | China | ARDL | PHH (✓) FDI increases both total $CO_2$ emissions and $CO_2$ emissions from solid fuel consumption. |
| Zheng and Sheng [33] | 1997–2009 | 30 Chinese provinces | OLS, GMM, RE, FE | PHH (✓) FDI increases $CO_2$ emissions. |
| Jiang et al. [34] | 2014 | 150 Chinese cities | Spatial analysis, OLS | No PHH. FDI reduces the air quality index. |
| Kathuria [35] | 2002–2010 | 21 Indian states | OLS, FE, RE | No PHH. |
| Liu et al. [36] | 2003–2014 | 285 Chinese cities | Spatial analysis | PHH (✓) for wastewater and sulfur dioxide emissions. No PHH for waste soot and dust. |
| Murthy and Gambhir [37] | 1991–2014 | India | OLS | PHH (✓) FDI increases $CO_2$ emissions. |
| Sung et al. [38] | 2000–2015 | 28 subsectors of the Chinese manufacturing sector | GMM | No PHH. FDI reduces $CO_2$ emissions. |
| Liu et al. [39] | 1996–2015 | China's 29 provinces | FE, RE | No PHH. N-shaped relationship between FDI and $CO_2$ emissions. |
| Rana and Sharma [40] | 1982–2013 | India | ARDL, ECM, TY-causality | PHH (✓) FDI increases $CO_2$ emissions. |

Note: ✓ shows the validity of the hypothesis.

To the best of our knowledge, no studies in the literature have taken into account hydropower and coal energy consumption while testing the PHH. Few studies have investigated the relationship between hydropower and environmental pollution. Solarin et al. [41] found that hydropower energy consumption exerts a negative effect on $CO_2$ emissions in China and India. Solarin et al. [42] obtained similar results in 80 developed and developing countries. Bello et al. [43] concluded that hydropower consumption reduces $CO_2$ emissions and carbon footprint in Malaysia. Destek and Aslan [44] found that hydropower usage reduces $CO_2$ emissions in Italy, the United Kingdom, and the United States (USA). Bildirici [45] concluded that hydropower increases environmental quality in Japan. In contrast to the findings of these studies, Pata [46] and Ummalla and Samal [3] reported that hydropower consumption has no impact on $CO_2$ emissions in Turkey and China, respectively. In line with these studies, Pata and Aydin [47] revealed that hydropower has no impact on the ecological footprint in the top six hydropower-consuming countries.

Studies examining the impact of coal consumption on environmental pollution are also limited. The studies that determined that coal consumption significantly increased $CO_2$ emissions include Tiwari et al. [48] for India, Shahbaz et al. [49] for South Africa,

Hao et al. [50] for 29 Chinese provinces, Pata [51] for Turkey, and Joshua and Bekun [52] for South Africa. In addition, two studies have comparatively examined the effects of coal consumption on environmental degradation in China and India. Shahbaz et al. [53] found that a 1% increase in coal consumption increased $CO_2$ emissions by 0.72% and 0.44% in China and India, respectively. Similarly, Farhani and Balsalobre-Lorente [54] argued that the negative impact of coal consumption on the environment is greater in China than in India. Contrary to the lack of consensus regarding the impact of FDI inflows and hydropower on the environment, the harmful effect of coal on environmental pollution has been determined in all studies.

Overall, there are three research gaps in the above literature. First, few studies have examined the PHH for China using the ARDL method [14]. Similarly, the number of studies conducted using this method are limited to India. Second, and most importantly, there are no studies in the literature that simultaneously test the impact of FDI, hydropower, and coal consumption on $CO_2$ emissions for China and India. Although hydropower accounts for 75% of global renewable energy resources, little attention has been paid to the environmental impacts of hydropower in the energy economics literature [43]. Third, the results may differ depending on the dependent variable that serves as an indicator of environmental pollution. For example, Liu [32] found that the PHH is not valid for $SO_2$ and soot emissions in China, while Sun et al. [16], Murthy and Gambhir [37], Rana and Sharma [40], and other researchers found that FDI increases $CO_2$ emissions. These findings may also differ depending on the data source. For this reason, we compiled $CO_2$ emissions from two different data sources in our study. We also used the carbon footprint to test the robustness of the results. To address these gaps and contribute to the existing literature, our study, for the first time, compares the impact of coal and hydropower on carbon pollution using the recently developed bootstrap ARDL approach.

### 3. Material and Methods

*3.1. Model and Data*

To analyse the validity of the PHH and the impact of energy consumption on carbon footprint and emissions in China and India, this study utilises the longest-available stretch of data covering the period from 1980 to 2016. In this study, the analysis period is limited to 37 observations, as Chinese data are available from 1980 onwards. With respect to three different carbon indicators, the study considers the following empirical models:

$$lnCF_t = \delta_0 + \delta_1 lnFDI_t + \delta_2 lnHEC_t + \delta_3 lnCC_t + \varepsilon_t \qquad (1)$$

$$lnCO_{2a_t} = \beta_0 + \beta_1 lnFDI_t + \beta_2 lnHEC_t + \beta_3 lnCC_t + z_t \qquad (2)$$

$$lnCO_{2b_t} = \vartheta_0 + \vartheta_1 lnFDI_t + \vartheta_2 lnHEC_t + \vartheta_3 lnCC_t + u_t \qquad (3)$$

where $t$ denotes the time period, $\delta_0$, $\beta_0$, and $\vartheta_0$ are the constant terms, $\delta_1$, $\delta_2$, $\delta_3$, $\beta_1$, $\beta_2$, $\beta_3$, $\vartheta_1$, $\vartheta_2$, and $\vartheta_3$ are the coefficients of the descriptive variables, and $\varepsilon_t$, $z_t$, and $u_t$ are the error terms. The dependent variables used in the study: $lnCF_t$ is the natural logarithm form of per capita carbon footprint (global hectares), $lnCO_{2a_t}$ illustrates the per capita $CO_2$ emissions (metric tons), and $lnCO_{2b_t}$ signifies per capita carbon dioxide emissions (kg oil equivalent) from a different source. These are explained by the following independent variables: $lnFDI_t$ defines per capita foreign direct investment (constant 2010 USD), $lnHEC_t$ refers to per capita hydropower energy consumption (kg oil equivalent), and $lnCC_t$ represents the per capita coal consumption (kg oil equivalent). The variables are retrieved from three different data sources. The $FDI_t$ and $CO_{2a_t}$ are collected from World Development Indicators (WDI) [55]. Further, $CC_t$ and $lnCO_{2b_t}$ data are derived from the British Petroleum Statistical Review of World Energy (BP) [56], while the $CF_t$ data are obtained from Global Footprint Network [57]. Table 2 provides detailed information on the variables.

**Table 2.** The definitions of variables used.

| Symbol | Variable | Definition | Measurement Unit |
|---|---|---|---|
| CF | Carbon footprint | CF is a type of ecological footprint in terms of carbon emissions from individual or mass production, consumption, and organizational activities. | global hectares per capita |
| $CO_{2a}$ | Carbon dioxide emissions from WDI | It includes carbon dioxide released by gas firing, cement production, and consumption of gaseous, liquid, and solid fuels. | metric tons per capita |
| $CO_{2b}$ | Carbon dioxide emissions from BP | It represents the emission of carbon into the atmosphere by each type of energy included in the IPCC emission factors list (For more information, please see https://www.ipcc-nggip.iges.or.jp/public/2006gl/pdf/2_Volume2/V2_0_Cover.pdf (accessed on 10 May 2021)). | kg oil equivalent per capita |
| FDI | Foreign direct investment | FDI is a cross-border investment made by a person into an institution or firm residing in another country. It represents direct investments in an economy as the sum of equity and earnings. | constant 2010 USD |
| HEC | Hydropower consumption | Hydropower is the use of water stored in reservoirs or drawn from flowing rivers to generate electricity. | kg oil equivalent per capita |
| CC | Coal consumption | The amount of coal burned for the purpose of electricity generation, industrial production, residential heating, and similar activities. | kg oil equivalent per capita |

In the above equations, if at least one of the $\delta_1$, $\beta_1$ or $\vartheta_1$ coefficients are positive and statistically significant, then the validity of the PHH is confirmed. Otherwise, if any of these coefficients are negative and statistically significant, countries can benefit from the pollution halo effect and thereby reduce pollution. The expected sign of the coefficients on coal consumption is positive [48,51,58]. However, the impact of hydropower on environmental pollution is not clear. On the one hand, Solarin et al. [41] argued that hydropower reduces $CO_2$ emissions in China and India. On the other hand, Pata [46], and Pata and Aydin [47] found that hydropower is not effective in reducing pollution. Although hydropower is a primary source of renewable energy, it can also harm the environment if hydropower reservoirs behave as potential sources of greenhouse gases rather than a carbon sink [59]. Therefore, there is controversy about the meaning and sign of the hydropower coefficient.

Figure 1 presents the level values of the variables made logarithmic. The level values of $CO_2$ emissions, carbon footprint, FDI, hydropower, and coal consumption have shown an increasing trend in both countries. It can also be seen that the per capita difference in FDI between China and India has decreased significantly since 2000. On the contrary, hydropower consumption differs between China and India over time. The descriptive statistics for the analysed variables are presented in Table 3.

Table 3 reveals that $lnCO_{2b}$ and lnCC have the highest mean and median values, while lnCF and $lnCO_{2a}$ have the lowest. Moreover, the skewness and kurtosis statistics indicate that all variables are skewed and leptokurtic. Besides, the probability of Jarque–Bera statistics show the null hypothesis of normality and cannot be rejected. Therefore, it is appropriate to use linear methods.

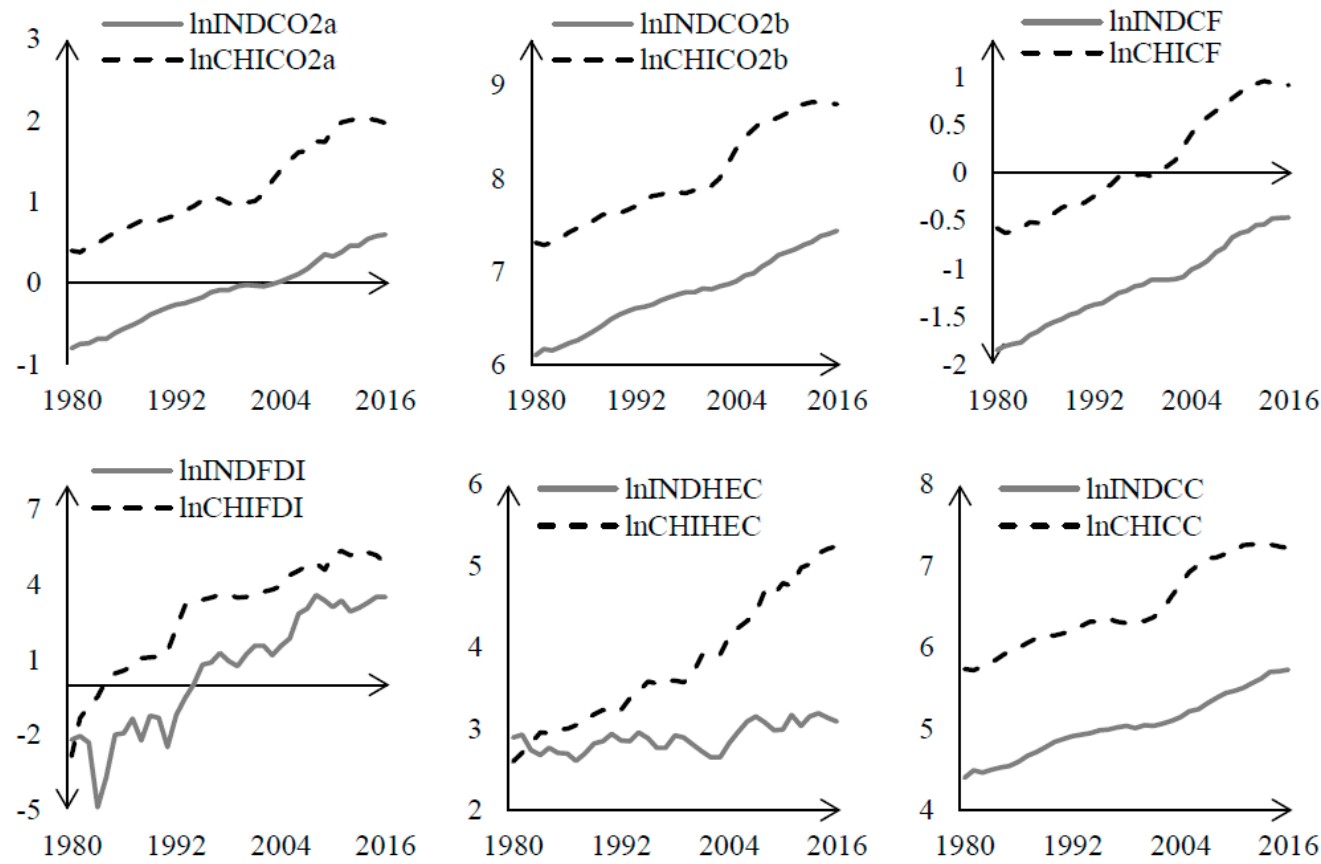

**Figure 1.** Visual representation of the variables.

**Table 3.** Descriptive statistics and normality properties.

| Country | Statistic | lnCF | lnCO$_{2a}$ | lnCO$_{2b}$ | lnFDI | lnHEC | lnCC |
|---|---|---|---|---|---|---|---|
| China | Mean | 0.113 | 1.161 | 8.026 | 2.854 | 3.802 | 6.497 |
| | Median | −0.029 | 1.008 | 7.867 | 3.506 | 3.584 | 6.320 |
| | Maximum | 0.961 | 2.022 | 8.821 | 5.367 | 5.243 | 7.279 |
| | Minimum | −0.623 | 0.378 | 7.284 | −2.845 | 2.597 | 5.718 |
| | Std. Dev | 0.546 | 0.533 | 0.525 | 2.174 | 0.792 | 0.521 |
| | Skewness | 0.303 | 0.382 | 0.312 | −0.805 | 0.406 | 0.273 |
| | Kurtosis | 1.665 | 1.821 | 1.653 | 2.695 | 1.922 | 1.688 |
| | Jarque–Bera | 3.112 | 3.041 | 3.396 | 4.146 | 2.806 | 3.112 |
| | *p*-value | (0.210) | (0.218) | (0.183) | (0.125) | (0.245) | (0.210) |
| India | Mean | −1.159 | −0.102 | 6.760 | 0.540 | 2.886 | 5.037 |
| | Median | −1.169 | −0.085 | 6.777 | 0.949 | 2.880 | 5.016 |
| | Maximum | −0.465 | 0.597 | 7.438 | 3.587 | 3.189 | 5.724 |
| | Minimum | −1.848 | −0.800 | 6.098 | −4.889 | 2.603 | 4.396 |
| | Std. Dev | 0.423 | 0.407 | 0.390 | 2.363 | 0.167 | 0.383 |
| | Skewness | 0.135 | 0.010 | 0.034 | −0.367 | 0.225 | 0.158 |
| | Kurtosis | 1.940 | 2.042 | 2.008 | 2.050 | 1.988 | 2.124 |
| | Jarque–Bera | 1.843 | 1.414 | 1.523 | 2.224 | 1.892 | 1.335 |
| | *p*-value | (0.397) | (0.493) | (0.466) | (0.328) | (0.388) | (0.512) |

Note: ( ) indicates probability values of Jarque–Bera statistics.

### 3.2. Method

This paper performs the bootstrap ARDL approach, which extends the conventional ARDL using the bootstrap resampling method. The conventional ARDL bounds testing approach, developed by Pesaran et al. [60], has been widely used in many studies to analyse long-term relationships. This approach has become quite popular as it allows the analysis

of time series with different orders of integration. However, there are some shortcomings and errors in the application of the ARDL bounds testing approach.

In the bounds testing approach, 2 different tests are applied: the F-test and the t-dependent test. Pesaran et al. [60] propose to compare the F-test with the critical bounds values. There are 2 different bounds to test the validity of the cointegration relationship. According to the order of integration of the variables, lower bounds I(0) and upper bounds I(1) critical values are used. If the calculated test statistic lies between the upper and lower bounds, the results are inconclusive. In the conventional ARDL approach, the null of no-cointegration should be rejected with both F- and t-dependent test statistics to discover the long-term relationship. In practice, many researchers use only the F-test and ignore the t-dependent test (e.g., [3,16]). In cases where the F-statistic is significant, but the coefficient of the lagged level-dependent variable is statistically insignificant, inaccurate results may be obtained [61]. Pesaran et al. [60] solved this problem by assuming that the dependent variable is I(1). However, this approach may not be an effective solution as traditional unit root tests suffer from low power and size properties. The lack of endogeneity is another problem with the traditional approach.

To overcome the above problems, McNown et al. [61] introduced the bootstrap ARDL procedure. In addition to the F- and t-dependent statistics, a new t-independent test was proposed in this approach. With the t-independent test, the significance of the independent variables is tested separately, eliminating the requirement that the dependent variable must be I(1). Concomitantly, more robust results can be obtained with the simultaneous application of F-overall, t-dependent, and t-independent tests [61]. In other words, cointegration, no cointegration, and degenerate cases can be more accurately determined using the bootstrap ARDL approach. As an added advantage, the method is suitable for models with more than 1 endogenous variable and small samples [62].

Moreover, the bootstrap ARDL approach has no lower or upper bounds. Since bootstrap critical values can give more accurate results than asymptotic ones, McNown et al. [61] generated critical values using bootstrap simulation. In doing so, they removed the instability where conventional ARDL test statistics lie between the lower and upper bounds. In summary, the bootstrap ARDL approach offers improvements in terms of size and power properties [63].

The bootstrap ARDL approach is based on the following unrestricted error correction model (UECM), as shown in Equation (4).

$$\Delta lnED_t = \delta_0 + \delta_1 \sum_{i=1}^{u} \Delta lnED_{t-i} + \delta_2 \sum_{i=0}^{k} \Delta lnFDI_{t-i} + \delta_3 \sum_{i=0}^{g} \Delta lnHEC_{t-i} + \delta_4 \sum_{i=0}^{d} \Delta lnCC_{t-i}$$
$$+ \varnothing_1 lnED_{t-1} + \varnothing_2 lnFDI_{t-1} + \varnothing_3 lnHEC_{t-1} + \varnothing_4 lnCC_{t-1} + v_t \tag{4}$$

where $\Delta$ is the difference operator; ED represents environmental degradation ($CO_2$ emissions or carbon footprint); $\vartheta_0$ illustrates the constant term; $u$, $k$, $g$, and $d$ are indices of optimal lags; $\delta_1$, $\delta_2$, $\delta_2$ and $\delta_4$ show the short-term dynamics; $\varnothing_1$, $\varnothing_2$, $\varnothing_3$ and $\varnothing_4$ demonstrate the long-term dynamics; $t$ represents the time periods $t = 1,2,3, \ldots ,T$; and $v_t$ is the error term with 0 mean and constant variance. In the Case III model (unrestricted constant and no trend), the following hypotheses are used to test the existence of cointegration:

- $H_0 : \varnothing_1 = \varnothing_2 = \varnothing_3 = \varnothing_4 = 0$ for F-overall test on all lagged level variables;
- $H_0 : \varnothing_1 = 0$ for t-dependent test on lagged level dependent variables;
- $H_0 : \varnothing_2 = \varnothing_3 = \varnothing_4 = 0$ for t-independent test on lagged level independent variables.

Based on these null hypotheses, we can define 2 non-cointegrating degenerate cases, and the presence or absence of cointegration as follows:

(1) Degenerate case #1: If the calculated F- and t-independent statistics are significant (the null hypotheses are rejected) but the t-dependent statistic is insignificant.
(2) Degenerate case #2: If the calculated F- and t-dependent statistics are significant but the t-independent statistic is insignificant.
(3) No cointegration: If all or at least 2 of the test statistics are insignificant.
(4) Cointegration: If all test statistics are significant at a minimum 5% level.

The cointegration status can be more clearly demonstrated using the new t-independent test [63]. The degenerate cases (1 and 2) show that there is no cointegration between the variables. For the existence of cointegration, the null hypothesis must be rejected with all 3 tests. To briefly explain the application of the bootstrap ARDL approach, a schematic flow is shown in Figure A1 (see Appendix A).

## 4. Results

If the variables are integrated at the first difference, various methods can be applied to investigate cointegration relationships ([64–66], among others). However, few methods such as bootstrap ARDL can be used when the variables have different orders of integration. Therefore, it is necessary to test the stationarity properties of the variables in order to decide which cointegration test is appropriate to investigate the relationships between carbon emissions and their determinants.

In the first stage of analysis, the study performs the conventional unit root tests of Dickey and Fuller (ADF) [67] and Phillips and Perron (PP) [68]. The study also applies the endogenous break unit root test of Zivot and Andrews (ZA) [69]. The findings of the unit root tests are presented in Table 4. On the one hand, the ADF and PP tests indicate that all variables have a unit root at level, except for Chinese FDI. On the other hand, the ZA test shows that Chinese HEC is stationary at level, while all other variables are stationary at first differences. Since the ZA unit root test accounts for an endogenous structural break, it is concluded that the HEC variable is stationary at level and the other variables are stationary at first difference. To summarise, the variables have a mixed order of integration, and therefore the bootstrap ARDL approach can be utilised to detect a cointegration relationship.

**Table 4.** Unit root test results.

| | Test | ADF | | PP | | ZA | | |
|---|---|---|---|---|---|---|---|---|
| Country | Variable | Level | First Dif. | Level | First Dif. | Level | First Dif. | Break Date |
| China | $\ln CF$ | −0.766 | −3.646 * | 0.076 | −3.703 * | −2.941 | −6.193 * | 2003 |
| | $\ln CO_{2a}$ | −0.910 | −3.466 ** | −0.234 | −3.357 ** | −4.156 | −6.179 * | 2008 |
| | $\ln CO_{2b}$ | −2.580 | −5.938 * | −2.580 | −6.081 * | −3.624 | −5.453 ** | 2003 |
| | $\ln FDI$ | −2.237 | −5.260 * | −4.260 * | - | −4.098 | −5.453 ** | 1992 |
| | $\ln HEC$ | 1.045 | −7.007 * | 1.271 | −7.032 * | −5.137 ** | - | 1998 |
| | $\ln CC$ | −1.477 | −1.824 | −0.564 | −2.494 | −4.362 | −5.782 * | 2009 |
| India | $\ln CF$ | −0.766 | −3.646 * | 0.076 | −3.703 * | −2.941 | −6.193 * | 1996 |
| | $\ln CO_{2a}$ | −0.910 | −3.466 * | −0.234 | −3.574 ** | −4.156 | −6.694 * | 2008 |
| | $\ln CO_{2b}$ | 0.076 | −2.107 | 0.021 | −6.167 * | −4.321 | −8.077 * | 1999 |
| | $\ln FDI$ | −0.865 | −6.331 * | −0.514 | −7.531 * | −4.192 * | −5.216 ** | 2006 |
| | $\ln HEC$ | −1.542 | −5.374 * | −1.665 | −5.439 * | −4.584 | −5.945 * | 1999 |
| | $\ln CC$ | 0.245 | −5.882 * | 0.153 | −5.966 * | −3.717 | −7.304 * | 1999 |

Note: The superscripts * and ** denote significance at 1% and 5% levels.

Moreover, considering the structural break dates determined in the ZA unit root test, the financial crisis (2008–2009) shows a significant impact on $CO_2$ emissions in both countries. The financial crisis also had an impact on the consumption of coal-based fossil fuels in China. With the 1992–1993 reform, China became one of the countries that attracted the most FDI [70]. Similarly, India experienced a massive surge in FDI inflows in 2006 with the removal of restrictions in the retail sector [71]. Furthermore, it is possible to explain the structural change in India's energy consumption and $CO_2$ emissions with the Kargil War in 1999. Due to these structural changes, it is more appropriate to determine the stationarity of the series with the ZA test results.

The next crucial step is to investigate the cointegration relationship between FDI, hydropower consumption and coal consumption and environmental degradation indicators. Table 5 reports the findings of the bootstrap ARDL cointegration test. The calculated test

statistics are compared with the bootstrapped critical values at the 1% and 5% significance levels. The results of the F-overall, t-dependent, and t-independent test statistics illustrate that the null hypothesis is rejected at the 5% significance level for Model 1 and 1% for Model 3 in China. However, no cointegration was confirmed for Model 2, as the F- and t-dependent statistics are insignificant. As for India, there is a cointegration relationship in Model 5, where $CO_2$ emissions compiled from the World Bank is the dependent variable. When $CO_2$ emissions from BP and carbon footprint are considered as dependent variables, the null hypothesis could not be rejected in Models 4 and 6.

**Table 5.** Cointegration analysis results.

| Panel (a): Test Statistics and Models | | | | | | |
|---|---|---|---|---|---|---|
| Country | Model | ARDL | F-Overall | *t*-Dep. | *t*-Indep. | Findings |
| China | (1)CF = $f$(FDI, HEC,CC) | 2,1,0,1 | 6.278 ** | −3.501 ** | 5.092 ** | Cointegrated |
| | (2)$CO_{2a}$ = $f$(FDI, HEC,CC) | 1,1,0,1 | 3.245 | −2.265 | 3.945 * | Non-cointegration |
| | (3)$CO_{2b}$ = $f$(FDI, HEC,CC) | 2,1,2,2 | 9.182 * | −4.437 * | 7.823 * | Cointegrated |
| India | (4)CF = $f$(FDI, HEC,CC) | 2,0,0,1 | 0.955 | −0.190 | 0.936 | Non-cointegration |
| | (5)$CO_{2a}$ = $f$(FDI, HEC,CC) | 1,0,0,0 | 4.874 ** | −3.615 * | 5.323 ** | Cointegrated |
| | (6)$CO_{2b}$ = $f$(FDI, HEC,CC) | 2,2,2,2 | 0.838 | −0.803 | 0.799 | Non-cointegration |

| Panel (b): Bootstrapped Critical Values for ARDL Procedure | | | | | | |
|---|---|---|---|---|---|---|
| Statistic | F-Overall | | *t*-Dep. | | *t*-Indep. | |
| Model | 1% | 5% | 1% | 5% | 1% | 5% |
| 1 | 6.401 | 4.396 | −3.924 | −2.773 | 4.029 | 3.043 |
| 2 | 5.721 | 3.604 | −3.454 | −2.633 | 5.967 | 3.941 |
| 3 | 6.234 | 4.248 | −3.356 | −2.467 | 6.684 | 3.974 |
| 4 | 4.866 | 3.337 | −3.266 | −2.368 | 5.484 | 3.686 |
| 5 | 6.687 | 4.266 | −3.439 | −2.676 | 7.969 | 5.298 |
| 6 | 5.472 | 3.518 | −2.715 | −1.832 | 5.823 | 3.660 |

Note: The superscripts * and ** denote significance at 1% and 5% levels. Optimal lags were selected using the AIC.

Consequently, the existence of cointegration is confirmed in three of the six models. Another important finding is that the cointegration relationship changes depending on the data source and type. On the one hand, with respect to $CO_2$ emissions, cointegration was confirmed in China with data from BP and in India with data from the World Bank. On the other hand, in terms of carbon footprint, there is cointegration between variables for China but not for India. Focusing on a single indicator may not accurately reflect the relationship between environmental degradation, FDI, and types of energy consumption. Therefore, it is useful to compare the results using different dependent variables.

In the last stage, the long- and short-term coefficients are estimated and reported in Table 6. For China, the coefficients are estimated by two different models. According to Model 1, the increase in FDI and hydropower consumption increases the carbon footprint in the short run. Keeping other things constant, a 1% increase in coal consumption increases the carbon footprint by 0.92%. The negative impact of these three variables on environmental quality continues in the long run. The long-term impact of coal consumption is less than the short term. Moreover, a 1% increase in hydropower consumption increases the carbon footprint by 0.12% in the short run and 0.18% in the long run. The relationship between FDI and carbon footprint is positive and statistically significant, implying that the PHH is valid for China. According to Model 3, although hydropower reduces $CO_2$ emissions in the short run, the effect of hydropower on environmental quality, in the long run, is negative. Coal consumption increases $CO_2$ emissions in both the short and long run. FDI increases environmental pollution in the short run but has no effect on $CO_2$ emissions in the long run. Therefore, the PHH is not valid for Model 3. This shows that the validity of the PHH is sensitive to the pollution indicator used for China.

**Table 6.** Short- and long-term coefficients.

| Panel (a): Long-Term Estimation | | | | | | |
|---|---|---|---|---|---|---|
| | Model 1 (CF, China) | | Model 3 ($CO_{2b}$, China) | | Model 5 ($CO_{2a}$, India) | |
| Variable | Coefficient | *t*-Stat. | Coefficient | *t*-Stat. | Coefficient | *t*-Stat. |
| lnFDI | 0.050 * | 3.498 | 0.036 | 1.630 | 0.018 ** | 2.346 |
| lnHEC | 0.178 ** | 2.229 | 0.143 ** | 2.225 | −0.091 | −1.291 |
| lnCC | 0.666 * | 5.750 | 0.747 * | 7.324 | 0.983 * | 16.859 |
| Constant | −5.029 * | −10.681 | 2.545 * | 5.927 | −4.769 * | −19.216 |
| **Panel (b): Short Term Estimation** | | | | | | |
| Variable | Coefficient | *t*-Stat. | Coefficient | *t*-Stat. | Coefficient | *t*-Stat. |
| $\Delta lnCF_{t-1}$ | −0.412 * | −2.931 | - | - | - | - |
| $\Delta lnCO_{2bt-1}$ | - | - | −0.427 * | −2.973 | - | - |
| $\Delta lnFDI$ | 0.043 * | 2.886 | 0.014 * | 2.904 | 0.005 | 1.130 |
| $\Delta lnHEC$ | 0.122 ** | 2.268 | 0.015 | 0.923 | −0.085 ** | −2.342 |
| $\Delta lnHEC_{t-1}$ | - | - | −0.036 ** | −2.446 | - | - |
| $\Delta lnCC$ | 0.921 * | 7.687 | 0.917 * | 23.568 | 0.568 * | 5.853 |
| $\Delta lnCC_{t-1}$ | - | - | 0.354 * | 2.822 | - | - |
| $\Delta$Constant | −2.308 * | −5.243 | 0.470 * | 6.119 | −2.526 * | −3.882 |
| $ECT_{t-1}$ | −0.458 * | −5.272 | −0.184 * | −6.034 | −0.523 * | −5.089 |
| **Panel (c): Diagnostic Check** | | | | | | |
| Test | LM | ARCH | White | BGP | Ramsey | Jarque–Bera |
| Model 1 | 0.175 (0.839) | 0.183 (0.671) | 0.468 (0.848) | 0.461 (0.853) | 0.015 (0.902) | 0.546 (0.760) |
| Model 3 | 0.906 (0.418) | 0.179 (0.674) | 1.039 (0.442) | 1.279 (0.295) | 0.019 (0.890) | 2.251 (0.324) |
| Model 5 | 0.939 (0.340) | 0.165 (0.686) | 0.256 (0.903) | 1.008 (0.418) | 1.692 (0.203) | 0.447 (0.799) |

Note: The superscripts * and ** denote significance at 1% and 5% levels. ( ) indicates probability values.

When it comes to India, FDI is shown to not affect $CO_2$ emissions in the short run. A 1% increase in hydropower consumption reduces pollution by 0.08%, while a 1% increase in coal consumption increases $CO_2$ emissions by 0.56%. In the short run, fossil-fuel-based energy consumption causes huge pollution in India. In the long run, coal consumption has a greater effect on increasing $CO_2$ emissions. A 1% increase in coal consumption increases $CO_2$ emissions by 0.98%, which identifies the importance of fossil fuel energy consumption on environmental quality. Hydropower has no impact on $CO_2$ emissions and is therefore ineffective in preventing environmental degradation. In addition to all these impacts, the positive and statistically significant coefficient of FDI implies that the PHH is valid for India.

The coefficients of the lagged error correction term ($ECT_{t-1}$) for all three models are negative and statistically significant at the 1% level, which supports the long-run relationship among hydropower, coal consumption, FDI, and carbon indicators. This also shows that imbalances in carbon indicators can be eliminated in the long run. The adjustment speed for deviations is about 2 years for $CO_2$ emissions in India, 5 years for $CO_2$ emissions and 2 years for carbon footprint in China. Moreover, all the estimated models have passed diagnostic tests such as the LM test for autocorrelation, ARCH, White, and BGP tests for heteroscedasticity of residuals, the Ramsey test for specification error, and the Jarque–Bera test for non-normality. Moreover, the CUSUM and CUSUMSQ tests are well within the 5% confidence lines, implying that the short- and long-run coefficients are stable (see Figure A2 in Appendix A).

In conclusion, the findings confirm the presence of the PHH in China and India. China is the largest coal consumer in the world. However, the negative impact of coal consumption on environmental quality is higher in India than in China. While hydropower has no effect on environmental quality in the long run in India, it increases $CO_2$ emissions

in China. The negative effect of hydropower is very limited compared to coal consumption. Therefore, hydropower can be substituted with coal for a cleaner environment. However, since hydropower does not reduce pollution in the long run, it is more important for China and India to promote the use of other types of renewable energy (e.g., solar, wind, and nuclear) and increase energy efficiency.

## 5. Discussion

This study concluded that coal consumption and FDI inflows increase carbon emissions in India and China. Moreover, the negative scale and composition effect of FDI inflows seem to prevail. The positive impact of FDI inflows on environmental degradation is consistent with the findings of He [31], Sun et al. [16], Zheng and Sheng [33], and Liu et al. [36] for China; and Acharyya [15], Murthy and Gambhir [37], and Rana and Sharma [40] for India; but contrasts with the findings of Zhang and Zhou [12], Liu et al. [32], Jiang et al. [34], Kathuria [35], Sung et al. [38], and Liu et al. [39], who argued that FDI improves environmental quality. As FDI inflows provide investment and employment opportunities in China and India, governments neglect environmental issues for economic development. The presence of the PHH shows that external funds are not used for environmental purposes in China and India. For this reason, countries should utilise their FDI inflows for environment-related research and development activities and provide tax exemptions, subsidies, and incentives to environmentally friendly foreign companies. In this way, the Chinese and Indian governments can attract foreign investors to prevent dirty investment.

Liu et al. [32] argued that both economic development and a clean environment could be provided simultaneously with FDI inflows, killing two birds with one stone. The authors identified this finding for $SO_2$ and soot emissions in China. In contrast, we argue that FDI has a negative impact on $CO_2$ emissions and carbon footprint. Therefore, FDI inflows do not provide a double win for China, both environmentally and economically.

China is experiencing a transformation from the secondary industrial sector to the service sector. Although the service sector accounts for more than 50% of the economy, the transportation sector has expanded unpredictably, bringing environmental problems [34]. This expansion of the transport sector has increased the demand for fossil fuels such as coal and oil. Therefore, the sectoral structure of FDI in China should be regulated and incentives should be provided to encourage investment in cleaner service sectors.

In terms of coal consumption, related studies with similar findings include Tiwari et al. [48], Shahbaz et al. [53], Hao et al. [50], and Farhani and Balsalobre-Lorente [54]. It is quite plausible that fossil fuel consumption causes high environmental pollution in developing countries. In 2019, China and India accounted for 51.7% and 11.8% of the world's total coal consumption, respectively [56]. In the same year, these two countries consumed nearly 100 EJ of coal, which is nine times the consumption of the USA. China reduced its share of coal consumption in the primary energy mix from 60% in 2017 to 58% in 2018, and although the share of coal consumption has fallen to historical levels, it is still the dominant energy source in China [72]. Similarly, India's coal consumption accounts for 56% of its primary energy mix. India is responsible for 70% of the 8.7% increase in global coal consumption in 2018 [73]. In both countries, reducing the share of coal in total energy consumption is a necessity to reduce carbon pollution. However, reducing coal consumption in China and India will have a negative impact on both industrial value added and economic growth. Implementing policies to reduce coal consumption alone can have negative impacts on the economies of China and India. Instead, improving efficiency in coal utilisation and replacing coal with renewable sources are more effective ways to reduce environmental pollution [53].

In addition, the results of our study revealed that hydropower is not one of the suitable RES to reduce carbon footprint and $CO_2$ emissions in China and India. Unlike the results of Solarin et al. [41,42], Bello et al. [43], Destek and Aslan [44], and Bildirici [45], our results are consistent with previous studies such as Pata [46], Ummalla and Samal [3],

and Pata and Aydin [47]. In China and India, other RES should be used to improve the environmental quality.

Our study has some limitations. First, this study does not include sectoral or regional data. The environmental impacts of coal and hydropower may vary by sector and region. Second, the effects of foreign trade, globalization, and similar important variables are neglected in the analysis due to methodological limitations. Third, the study focuses on carbon emissions and thus only one aspect of environmental problems: air pollution. With other methods, data, and environmental indicators, these limitations can be removed in future studies. As another proposal, the impact of hydropower and coal on environmental pollution in developed countries can be analysed under the PHH. Moreover, the factors affecting carbon footprint and $CO_2$ emissions in developed and developing countries can be studied simultaneously using current and advanced time series methods.

## 6. Conclusions

Especially since the 1980s, the increasing volume of foreign trade, international capital flows and fossil fuel consumption has led to social, economic, and environmental changes. The FDI influx, in particular, has spurred economic growth in developing countries such as China and India. However, economic expansion, energy consumption, and investment have increased environmental pollution in developing countries. Developed countries such as Germany and the United States have begun to reduce carbon emissions by shifting some of their polluting production to developing countries. Although this FDI flow and the resulting fossil fuel energy consumption bring economic benefits to developing countries, it may cause irreversible environmental problems in the future. Therefore, developing countries need to attract clean FDI and substitute renewable energy sources instead of coal, which has the highest carbon content. In this context, the main objective of this study is to analyse the impact of FDI, hydropower and coal consumption on carbon pollution using the PHH in China and India for the period 1980–2016. For this purpose, three indicators representing carbon pollution are used in the study.

The empirical findings confirm the existence of cointegration between FDI, hydropower, coal consumption, and various carbon indicators. The empirical results demonstrate that a cointegration relationship exists between the variables in three of the six models (for carbon footprint and $CO_2$ emissions from BP in China and $CO_2$ emissions from the World Bank in India). This shows that the cointegration relationship varies from country to country depending on the data source of $CO_2$ emissions and carbon footprint. The estimated coefficients show that hydropower consumption increases $CO_2$ emissions and the carbon footprint in China while it has no impact on $CO_2$ emissions in India. Moreover, coal consumption causes an enormous increase in environmental degradation in both countries. Furthermore, FDI inflows stimulate environmental degradation in China for carbon footprint and in India for $CO_2$ emissions. Thus, the results empirically validate the PHH. It is concluded that FDI and coal play a significant role in carbon pollution for both countries while hydropower consumption does not reduce carbon footprint and $CO_2$ emissions.

Regarding the policy implications of this study, the findings suggest that reducing coal consumption in China and India is of utmost importance for a better environment. Since coal consumption puts tremendous pressure on the environment, policymakers and the private sector should turn to the use of alternative "clean" energy sources. In this regard, China and India are among the top hydropower-consuming countries in the world. However, according to the results of the analysis, hydropower energy consumption in China has a positive effect on carbon footprint, while in India it does not affect $CO_2$ emissions. The pollution-increasing effect of hydropower consumption is quite small compared to coal in China. Nevertheless, hydropower is not an effective means of pollution control in these countries because it not only consumes high reserves and land but also alters the flow of rivers and lakes. In terms of energy policy, the governments of China and India should reduce energy intensity, increase energy efficiency, and promote the use of more environmentally friendly energy sources from solar and wind.

In addition, as FDI contributes to the rising $CO_2$ emissions in both countries, policymakers should introduce regulatory measures to prevent the negative impact of FDI inflow on environmental quality. In China and India, FDI inflows are managed without a proper environmental management system. To solve this problem, on the one hand, the transfer of environmental technologies of foreign enterprises should be accompanied by FDI inflows, and these technologies should be adapted to the production process of Chinese and Indian companies. On the other hand, the governments of China and India should optimize the FDI structure with environmental concerns in mind. Meanwhile, shifting FDI to energy-saving production instead of polluting industrial production can contribute to the development of environmental quality.

The results of the study offer some lessons for other developing countries. Coal consumption is the largest contributor to carbon emissions in China and India. Hydropower is also not very effective at preventing environmental problems. In light of this context, developing countries whose coal consumption is not yet as high as in China and India should promote renewable energy sources such as solar, wind, and biomass. To achieve this, the governments of developing countries can use a range of policy tools such as Renewable Portfolio Standards, tax exemptions for companies that generate electricity from renewable sources, and the spread of educational programs that raise people's awareness of clean energy utilisation. It is possible to prevent irreversible environmental problems and global warming if developing countries take action to improve environmental quality before it is too late.

**Author Contributions:** U.K.P.: data curation, methodology investigation, formal analysis, writing—original draft, writing—review and editing. A.K.: conceptualization, resources, writing—original draft, review and editing. All authors have read and agreed to the published version of the manuscript.

**Funding:** This research received no external funding.

**Institutional Review Board Statement:** Not applicable.

**Informed Consent Statement:** Not applicable.

**Data Availability Statement:** The data presented in this study are available on request from the first author.

**Acknowledgments:** We are thankful for the constructive and helpful comments made by two anonymous reviewers.

**Conflicts of Interest:** The authors declare no conflict of interest.

## Abbreviations

Autoregressive distributed lag: ARDL; British Petroleum: BP; carbon dioxide: $CO_2$; error correction model: ECM; exajoules: EJ; fixed effect estimator: FE; foreign direct investment: FDI; generalized method of moments: GMM; gigawatt: GW; international energy agency: IEA; Lagrange Multiplier: LM; Middle East and North Africa: MENA; million tonnes of oil equivalent: mtoe; organisation for economic co-operation and development: OECD; ordinary least square: OLS; pollution haven hypothesis: PHH; random effect estimator: RE; renewable energy sources: RES; stochastic impacts by regression on population affluence, and technology model: STIRPAT; sulfur dioxide: $SO_2$; Toda-Yamamoto: TY; world development indicators: WDI.

## Appendix A

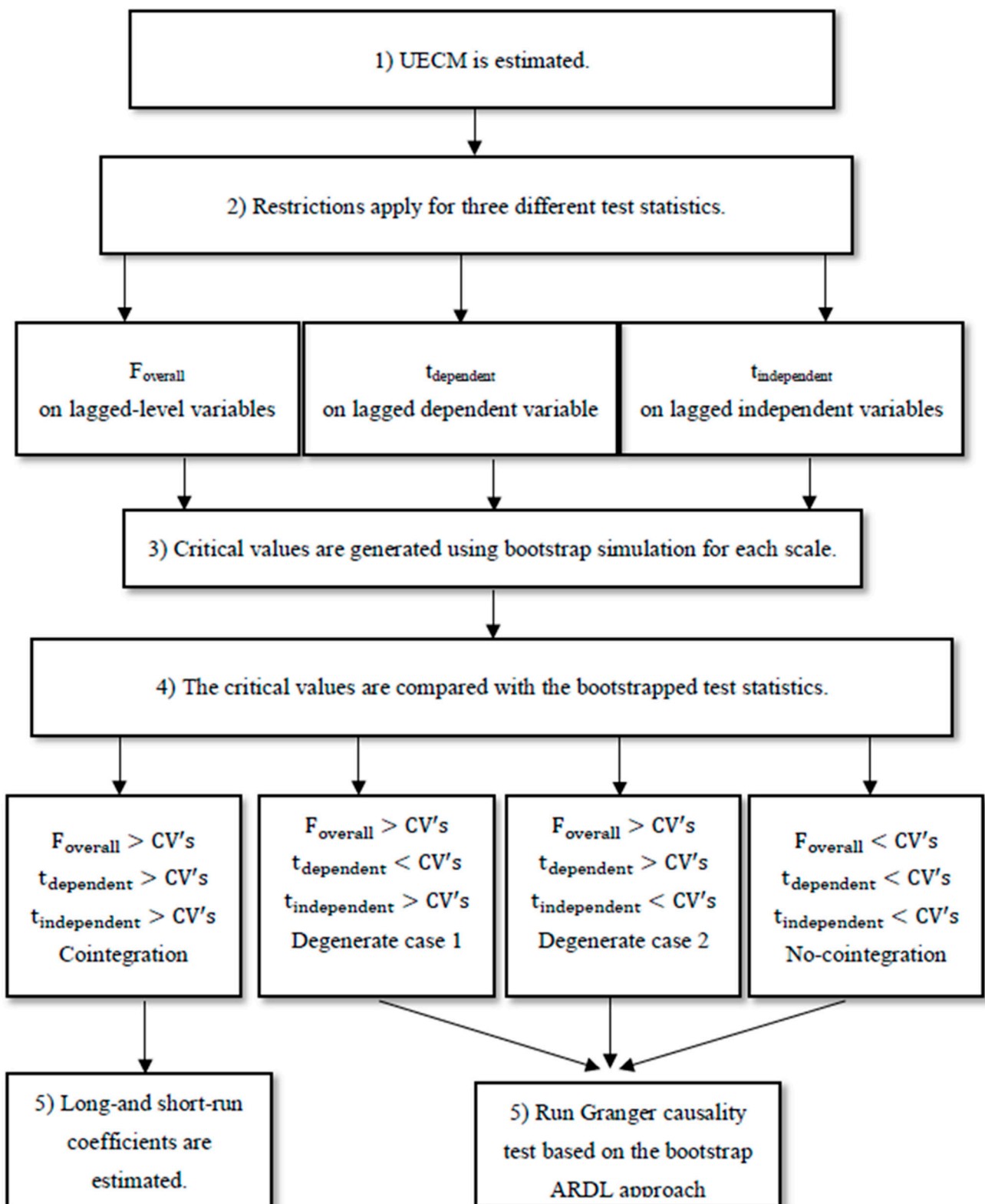

**Figure A1.** A. flow chart of the stages of the bootstrap ARDL approach.

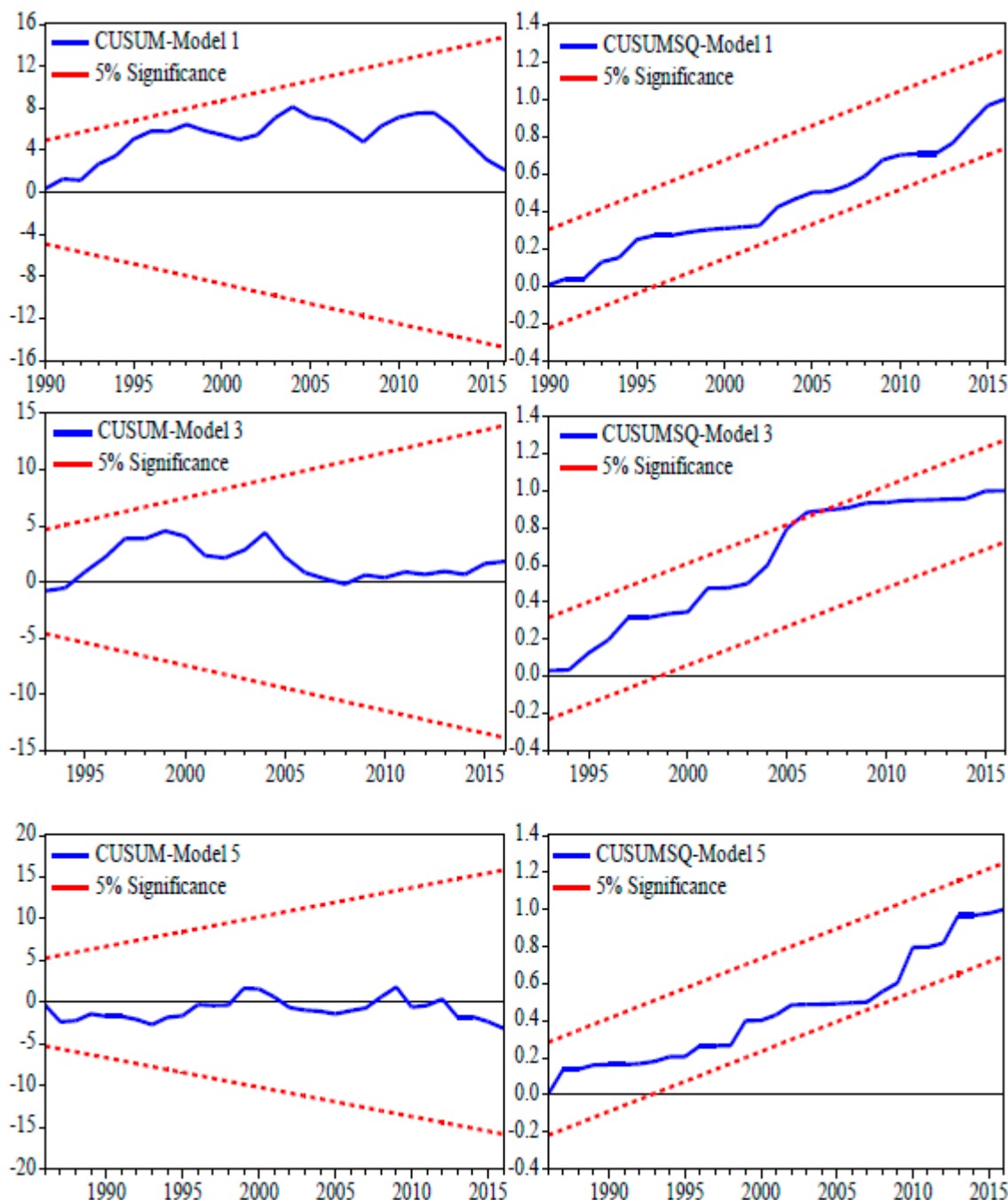

**Figure A2.** A. CUSUM and CUSUMSQ results.

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
