# Peer review of "The Influence of Hydropower and Coal Consumption on Greenhouse Gas Emissions: A Comparison between China and India"

_water, doi:10.3390/w13101387_

Round 1

Reviewer 1 Report

This is a very interesting study. I enjoyed reading the manuscript. Nevertheless, it needs some further improvements. In general, there are still some occasional grammar errors throughout the manuscript, especially the article "the," "a," and "an" is missing in many places; please make a spellchecking in addition to these minor issues. The reviewer has listed some specific comments that might help the authors further enhance the manuscript's quality. 1. Specific Comments A list of acronyms is needed. • Introduction • The objectives should be more explicitly stated. • Please elaborate a bit more the introduction section regarding the importance of hydropower plants development in achieving the sustainability goals. In this regard, the following literature may be helpful to: >, >, you may consider additional references as well. • What is the novelty of this work? • Methods • The methodology limitation should be mentioned. • All variables should be explained. • Results • This section is well written. • North arrow and scale bare missing in figures 3 and 4. • Discussion • The discussion should summarize the main finding(s) of the manuscript in the context of the broader scientific literature and address any limitations of the study or results that conflict with other published work.

Author Response

Response to Reviewer 1 Comments

Point 1 √:  In general, there are still some occasional grammar errors throughout the manuscript, especially the article "the," "a," and "an" is missing in many places; please make a spellchecking in addition to these minor issues.

Response 1: We edited minor grammatical and spelling mistakes in the text, and highlighted these edits in red. Thank you for your contribution.

Point 2 √: A list of acronyms is needed.

Response 2: We did not usually see a list of abbreviations in MDPI journals. As a result of our research, we found that Abbreviations is included in the following article.

https://www.mdpi.com/2071-1050/13/9/4995

For this reason, we have included an Abbreviations list after the abstract, based on your suggestion.

Abbreviations: autoregressive distributed lag: ARDL; British Petroleum: BP; carbon dioxide: CO2; error correction model: ECM; fixed effect estimator: FE; foreign direct investment: FDI; generalized method of moments: GMM; gigawatt: GW; international energy agency: IEA; Lagrange Multiplier: LM; Middle East and North Africa: MENA; million tonnes of oil equivalent: mtoe; organisation for economic co-operation and development: OECD; ordinary least square: OLS; pollution haven hypothesis: PHH; random effect estimator: RE; renewable energy sources: RES; stochastic impacts by regression on population affluence, and technology model: STIRPAT; sulfur dioxide: SO2; Toda-Yamamoto: TY; world development indicators: WDI.

√Point 3:  • Introduction • The objectives should be more explicitly stated.

  • Please elaborate a bit more the introduction section regarding the importance of hydropower plants development in achieving the sustainability goals. In this regard, the following literature may be helpful to: >, >, you may consider additional references as well. 

Response 3: We stated the purpose of the study more explicitly in the introduction.

To date, there has been no comparative analysis on the environmental impact of hydropower and coal consumption for China and India. These two countries are responsible for more than 35% of the global population. In addition, China and India are the two largest coal consumers and are among the top six largest hydropower consuming countries in the world. This high population and coal consumption cause various environmental problems such as increased CO2 emissions. China and India are two of the three highest CO2 emitting countries in the world, and therefore reducing carbon emissions is important both nationally and globally. In this regard, this study aims to examine the long-run impacts of hydropower, coal consumption and FDI on carbon-based pollution for India and China using an advanced empirical model.

We explained the contribution of hydropower to sustainable development.

As a low-carbon, low installation cost, reliable and clean source of energy, hydropower is playing an important role in carbon reduction and further helps in mitigating future climate change. Compared with highly volatile and non-steady renewable energy, flexible hydropower can be highly supportive to secure reliable grid operations [1].

  1. Sun, L.; Niu, D.; Wang, K.; Xu, X. Sustainable development pathways of hydropower in China: Interdisciplinary qualitative analysis and scenario-based system dynamics quantitative modeling. Journal of Cleaner Production 2021 287, 125528. https://doi.org/10.1016/j.jclepro.2020.125528

√Point 4:  What is the novelty of this work? 

Response 4: Our study is the first study to examine the effect of hydropower and coal on carbon pollution using bootstrap ARDL method. The novelty of our study is the current time series method. Moreover, our study differs from other studies by simultaneously analyzing the effect of hydropower and coal consumption on carbon pollution. We stated this as follows:

To address these gaps and contribute to the existing literature, our study, for the first time, compares the impact of coal and hydropower on carbon pollution using the recently developed bootstrap ARDL approach.

√ Point 5: The methodology limitation should be mentioned. 

Response 5: We used the Bootstrap ARDL method because the variables are stationary at different order of integration. There is only one constraint on the method. The code we've implemented allows only three independent variable. We have expressed this limitation as follows.

Second, the effects of foreign trade, globalization, and similar important variables are neglected in the analysis due to methodological limitations.

Point 6 √:  All variables should be explained.

Response 6: We have added detailed explanations of the variables with Table 2. Thank you for your contribution.

Table 2. The definitions of the variables

Symbols

Variables

Definition

Measurement unit

CF

Carbon footprint

CF is a type of ecological footprint in terms of carbon emissions from individual or mass production, consumption and organizational activities.

global hectares per capita

CO2a

Carbon dioxide emissions from WDI

It includes carbon dioxide released by gas firing, cement production, and consumption of gaseous, liquid, and solid fuels.

metric tons per capita

CO2b

Carbon dioxide emissions from BP

It represents the emission of carbon into the atmosphere by each type of energy included in the IPCC emission factors list[1].

kg oil equivalent per capita

FDI

Foreign direct investment

FDI is a cross-border investment made by a person into an institution or firm residing in another country. It represents direct investments in an economy as the sum of equity and earnings.

constant 2010 US$

HEC

Hydropower consumption

Hydropower is the use of water stored in reservoirs or drawn from flowing rivers to generate electricity.

kg oil equivalent per capita

CC

Coal consumption

The amount of coal burned for the purpose of electricity generation, industrial production, residential heating and similar activities.

kg oil equivalent per capita

Point 7: North arrow and scale bare missing in figures 3 and 4.

Response 7: We reviewed the figures and added arrows to them. Thank you

√ Point 8: The discussion should summarize the main finding(s) of the manuscript in the context of the broader scientific literature and address any limitations of the study or results that conflict with other published work.

Response 8:

In the discussion part of our study, the main findings about FDI, coal and HEC are presented, respectively.

This study concluded that coal consumption and FDI inflows increase carbon emissions in India and China

In terms of coal consumption, related studies with similar findings include Tiwari et al. [48], Shahbaz et al. [53], Hao et al. [50], and Farhani and Balsalobre-Lorente [54]. It is quite plausible that fossil fuel consumption causes high environmental pollution in developing countries.

In addition, the results of our study revealed that hydropower is not a suitable RES to reduce carbon footprint and CO2 emissions

Our findings of conflict with other studies are included in the discussion part of our study.

but contrasts with the findings of Zhang and Zhou [12], Liu et al. [32], Jiang et al. [34], Kathuria [35], Sung et al. [38], and Liu et al. [39]

Unlike the results of Solarin et al. [41,42], Bello et al. [43], Destek and Aslan [44], and Bildirici [45],

**At the end of the discussion part, we have stated three main constraints of our work on your proposal. Thank you for your valuable contribution.

Our study has some limitations. First, this study does not include sectoral or re-gional data. The environmental impacts of coal and hydropower may vary by sector and region. Second, the effects of foreign trade, globalization, and similar important variables are neglected in the analysis due to methodological limitations. Third, the study focuses on carbon emissions and thus only one aspect of environmental problems, air pollution. With other methods, data and environmental indicators, these limitations can be removed in future studies.

Reviewer 2 Report

The main objective of the study was to analyze the impact of FDI, hydropower, and coal consumption on carbon pollution using the PHH hypothesis in China and India for the period 1980-2016. In this regard, three indicators representing carbon pollution were used in the study. Remarks: What are the reasons for comparison India and China? It should be explained in detail. Line 104-107: More detailed information about the structure of the paper should be presented. The beginning of the conclusions seems like abstract. The more attractive article could concern only China with other leading carbon emitters. Please, present the conflicting views in the literature on predicting the impact of FDI on environmental pollution under the PHH in comparison to the presented approach. What lessons should other developing countries draw from this analysis? Some guidelines about the practical use of the obtained results should be presented. The last point of the article contains in fact only the conclusions relating to the researched case study, but there is no more detailed perspective. Include the digital object identifier (DOI) for all references where available.

Author Response

Response to Reviewer 2 Comments

Point 1: What are the reasons for comparison India and China? It should be explained in detail. 

Response 1: We have stated the reasons in the paragraph below. Thank you for your suggestion.

To date, there has been no comparative analysis on the environmental impact of hydropower and coal consumption for China and India. These two countries are responsible for more than 35% of the global population. In addition, China and India are the two largest coal consumers and are among the top six largest hydropower consuming countries in the world. This high population and coal consumption cause various environmental problems such as increased CO2 emissions. China and India are two of the three highest CO2 emitting countries in the world, and therefore reducing carbon emissions is important both nationally and globally. In this regard, this study aims to examine the long-run impacts of hydropower, coal consumption and FDI on carbon-based pollution for India and China using an advanced empirical model.

√ Point 2: Line 104-107: More detailed information about the structure of the paper should be presented.

Response 2: In accordance with your suggestion, we revised and developed the relevant part as follows. Thank you for your contribution.

The remainder of the paper is organised as follows: Section 2 reviews the findings of studies testing the PHH hypothesis for China and India. Section 3 defines different models for three pollution indicators used in the study and explains the application and advantages of the bootstrap ARDL method. Section 4 evaluates the results of the unit root test, cointegration relations, and the short and long run estimates. Section 5 discusses the empirical findings and limitations of this study. Finally, the last section presents policy recommendations and conclusions.

√ Point 3: The beginning of the conclusions seems like abstract. The more attractive article could concern only China with other leading carbon emitters. 

Response 3: We made the introduction of the conclusion of the study more attractive. Thank you for your valuable contribution.

Especially since the 1980s, the increasing volume of foreign trade, international capital flows and fossil fuels consumption have led to social, economic and environmental changes. The FDI influx, in particular, has spurred economic growth in developing countries such as China and India. However, economic expansion, energy consumption, and investment have increased the environmental pollution in developing countries. Developed countries such as Germany and the United States have begun to reduce carbon emissions by shifting some of their polluting production to developing countries. Although this FDI flow and the resulting fossil fuel energy consumption brings economic benefits to developing countries, it may cause irreversible environmental problems in the future. Therefore, it is important for developing countries to attract clean FDI and substitute renewable energy sources instead of coal, which has the highest carbon content. In this context, the main objective of this study is to analyze the impact of FDI, hydropower and coal consumption on carbon pollution using the PHH hypothesis in China and India for the period 1980-2016. For this purpose, three indicators representing carbon pollution are used in the study.

√ Point 4: Please, present the conflicting views in the literature on predicting the impact of FDI on environmental pollution under the PHH in comparison to the presented approach.

Response 4: We deleted the following sentence in the introduction.

Moreover, there are conflicting views in the literature on predicting the effects of FDI on environmental pollution under the PHH.

In the literature section, we have given the findings related to PHH in the table. We have stated the conflicting views about the findings with the following sentence. Thanks for your suggestion.

As can be seen in the table, there is no consensus on the validity of the PHH hypothesis for China and India. Four of the nine studies in the literature argue that the PHH is valid in China, while the other five discuss the opposite results. For India, one out of four studies does not support the PHH. It is expected that the present study help in providing insights for sustainable hydro-power future so that both countries can achieve their emissions target, and contribute to carbon reduction and global warming impacts.

√ Point 5: What lessons should other developing countries draw from this analysis? Some guidelines about the practical use of the obtained results should be presented. The last point of the article contains in fact only the conclusions relating to the researched case study, but there is no more detailed perspective.

Response 5: Based on your suggestion, we have indicated the lessons that other countries can learn from the findings for China and India. In addition, we presented the incentives for the use of renewable energy with policy recommendations in a more detailed perspective. In this way, the conclusion part of the study is much better. We are grateful for your constructive contribution

The results of the study offer some lessons for other developing countries. Coal consumption is the largest contributor to carbon emissions in China and India. Hydropower is also not very effective in preventing environmental problems. In light of this context, developing countries whose coal consumption is not yet as high as in China and India should promote renewable energy sources such as solar, wind and biomass. To do this, the governments of developing countries can use a range of policy tools, such as Renewable Portfolio Standards, tax exemptions for companies that generate electricity from renewable sources, and the spread of educational programs that raise people's awareness of clean energy utilization. It will be possible to prevent irreversible environmental problems and global warming if developing countries take action to improve environmental quality before it is too late.

√ Point 6: Include the digital object identifier (DOI) for all references where available.

Response 6: We added DOI information to all references.

Round 2

Reviewer 2 Report

Accept in present form.